# Responses of Physiology, Photosynthesis, and Related Genes to Saline Stress in *Cornus hongkongensis* subsp. *tonkinensis* (W. P. Fang) Q. Y. Xiang

**DOI:** 10.3390/plants11070940

**Published:** 2022-03-30

**Authors:** Jia-Qiu Yuan, Da-Wei Sun, Qiang Lu, Ling Yang, Hao-Wei Wang, Xiang-Xiang Fu

**Affiliations:** 1Co-Innovation Center for Sustainable Forestry in Southern China, Nanjing Forestry University, Nanjing 210037, China; jq.yuan1001@gmail.com (J.-Q.Y.); sundw1119@163.com (D.-W.S.); qianglu@njfu.edu.cn (Q.L.); 2College of Forestry, Nanjing Forestry University, Nanjing 210037, China; 3Shanghai Wildlife and Protected Natural Areas Research Center, Shanghai 200336, China; 15205185627@163.com; 4Suzhou Forestry Station, Suzhou 215128, China; whw1743702523@163.com

**Keywords:** *Cornus hongkongensis* subsp. *tonkinensis*, osmotic regulators, photorespiration, photosynthesis, salinity tolerance

## Abstract

*Cornus hongkongensis* subsp. *tonkinensis* (W. P. Fang) Q. Y. Xiang is a native evergreen species with high ornamental value for abundant variations in leaf, bract, fruit, and tree gesture. To broaden its cultivation in coastal saline soil, salt damage and survival rate, physiological responses, photosynthetic performance, and related genes were evaluated for annual seedlings exposed to 0.3% salt (ST) concentrations for 60 days. Syndromes of salt damage were aggravated, and the survival rate decreased with prolonged stress duration; all stressed seedlings displayed salt damage, and 58.3% survived. Under short-term saline stress (5 d), marked increases in malondialdehyde (MDA), relative electrical conductivity (REC), and decreases in superoxide dismutase (SOD), photosynthetic rate (P_n_), stomatal conductance (gs), and internal carbon dioxide concentration (C_i_) were recorded. The stable leaf water use efficiency (WUE) and chlorophyll content were positive physiological responses to ensure photosynthetic performance. Meanwhile, the expression levels of genes related to photosystem II (*psbA*) and photorespiration (*SGAT* and *GGAT*) were upregulated, indicating the role of photorespiration in protecting photosynthesis from photoinhibition. After 30 days of stress (≥30 d), there was a significant increase in MDA, REC, soluble sugar (SS), soluble protein (SP), and C_i_, whereas descending patterns in P_n_, gs, WUE, the maximal photochemical efficiency of photosystem II (F_v_/F_m_), and potential activities of PSII (F_v_/F_0_) occurred in salt-stressed seedlings, compared with CK. Meanwhile, the expression levels of related genes significantly dropped, such as *psbA*, *LFNR*, *GGAT*, *GLYK*, and *PGK*, indicating photoinhibition and worse photosynthetic performance. Our results suggest that the moderate salt tolerance of *C. hongkongensis* subsp. *tonkinensis* mostly lies in a better photosynthetic system influenced by active photorespiration. Hence, these results provide a framework for better understanding the photosynthetic responses of *C. hongkongensis* subsp. *tonkinensis* to salt stress.

## 1. Introduction

Continuous salinization of arable lands has already had a devastating impact with an annual rate of 10% forecast, and approximately 50% of the total cultivated land area worldwide will be salinized by 2050 [1]. Salt stress, which induces major damage in plants, is one of the most important abiotic stresses for plants. Osmotic stress was due to water and nutrient uptake inhibition from roots, and ion toxicity was due to oxidative stress caused by excessive sodium entrance—these results in unbalanced photosynthesis and metabolism [2]. To survive in salinity conditions, plants develop major physiological response mechanisms as follows: (1) selective absorption, transportation, and isolation of toxic ions; (2) enhancement of the activity of antioxidant enzymes (e.g., superoxide dismutase SOD, catalase CAT, etc. [3,4]) and synthesis of nonenzymatic antioxidants to allow ROS scavenging; (3) accumulation of organic (Na^+^, K^+^ and Cl^−^) or inorganic (soluble sugar SS, soluble protein SP, proline Pro, etc.) [3] solutes to reduce osmotic potential, and (4) adjustment of the hormonal profile to cope with salt stress [5].

Ordinarily, photosynthesis has been recognized as an important pathway for providing energy to plants in growth and development. Many studies have concluded that the reduction in the photosynthesis rate (P_n_) under NaCl stress is a consequence of several physiological responses, including lower stomatal conductance (gs), transpiration rate (T_r_), chlorophyll, and carotenoid contents, or a combination of these parameters [6]. Notably, prolonged salt stress can damage the photosynthetic apparatus, particularly PSII, resulting in photoinhibition [6,7]. More harmful reactive oxygen species (ROS) have been produced in PSII and PSI while photoinhibition occurred [8]. Nevertheless, plants try to protect PSII against photoinhibition by rapid turnover of the D1 protein of PSII, encoded by the *psbA* gene [9]. Ferredoxin-NADP^+^ oxidoreductase (FNR), participating in cyclic electron flow around PSI, is also involved in oxidative stress responses of higher plants; the production of ROS results in a marked release of FNR, which might aim to maintain the NADP^+^/NADPH homeostasis in stressed plants [10]. Nowadays, photorespiration is acknowledged to be a protective mechanism against photoinhibition, which contributes to the protection of PSII from oxidative stress and PSI by optimizing its redox state [11]. Among them, serine-glyoxylate aminotransferase SGAT (EC 2.6.1.45) and glutamate-glyoxylate aminotransferase GGAT (EC 2.6.1.4) are two key enzymes in photorespiration that localize in peroxisomes [12,13]. It has been reported that increased SGAT activity in salt-stressed transgenic duckweed resulted in a higher maximum quantum yield of photosystem II (F_v_/F_m_), improved cell membrane integrity, and a strengthened antioxidant system [14]. GGAT may inhibit chlorophyll degradation or accelerate N metabolism to improve the saline stress tolerance of plants [12,15]. Another key enzyme-like glutamate synthase (GOGAT) could promote the generation of osmoprotective amino acids, such as proline [16]. Moreover, another important function of photorespiration is to drive the Calvin cycle under a limited CO_2_ supply [17]. A relevant study showed that higher expression levels of FBPaldolase, FBPase, and SBPase increased the RuBisCO regeneration to cope with stress [18]. Although the possible physiological and photosynthetic responses to salt stress have been well studied, the mechanisms in various plant species are divergent [19].

Trees in *Cornus*, belonging to the family Cornaceae, are a cluster of small evergreen or deciduous species with high ornamental value for showy bracts, colorful leaves, and fruit [20]. Two groups in *Cornus* include the East Asia group and the North America group [21]. Eastern Asian dogwoods, native to subtropical areas in China, possess stronger resistance to drought, heat, and disease stress than North American ones [7,22,23,24]. Field trials revealed that some dogwoods of Eastern Asian (i.e., *C. elliptica* (Pojarkova) Q. Y. Xiang & Boufford, *C. hongkongensis* subsp. *elegans* (W. P. Fang et Y. T. Hsieh) Q. Y. Xiang, *C.* subsp. *tonkinensis* (W. P. Fang) Q. Y. Xiang) grow well on the saline soil with 0.11–0.21% salt content in Dafeng Forest Farm, Jiangsu Province. Additionally, it is reported that half seedlings of *C. florida* L. and *C. hongkongensis* subsp. *elegans* under 0.3% salt stress could survive [3,7]. Thereinto, *C. hongkongensis* subsp. *tonkinensis* is considered a potential species for further development owing to its rich leaf color variation and good resistance to adversity. Expectedly, it is recommended as the preferred rootstock for cultivars originating from the North American group.

Thus, our aim in this study was to expound on the photosynthetic response of salt-tolerant plants by analyzing photosynthetic traits and related genes in salt-stressed seedlings of *C. hongkongensis* subsp. *tonkinensis*. Our study highlights adaptive changes in photosynthesis and provides new evidence on how salt-stressed seedlings relieve photoinhibition by adjusting photorespiration.

## 2. Materials and Methods

### 2.1. Plant Materials, Growth Conditions, and Experimental Design

The present study was carried out in a half-open greenhouse at Nanjing Forestry University, China, from June to August 2019. One-year-old seedlings with uniform size were transplanted from containers into a hydroponic tank (50 L) filled with half-strength modified Hoagland solution (2.50 mM Ca(NO_3_)_2_·4H_2_O; 2.50 mM KNO_3_; 0.50 mM NH_4_NO_3_; 1.00 mM MgSO_4_·7H_2_O; 0.25 mM KH_2_PO_4_; 0.05 mM Fe-EDTA; 23.13 μM H_3_BO_3_; 4.57 μM MnCl_2_·4H_2_O; 0.38 μM ZnSO_4_·7H_2_O; 0.10 μM CuSO_4_·5H_2_O; 0.28 μM H_2_MoO_4_·H_2_O, pH 6.0), and an aeration device was used to ensure enough fresh air for root breathing. The nutrient solution was renewed weekly to maintain a stable growth environment. The adaptive culture in the above solution lasted three months until the formation of developed leaves.

Subsequently, the nutrient solution was added with coarse sea salt from the Yellow Sea (to simulate the coastal saline environment of the Yangtze–Huaihe Region), which is mainly composed of chlorine, sodium, sulfur, magnesium, calcium, potassium, carbon, bromine, strontium, boron, and fluorine. Two salt concentrations were used as treatments: 0(CK) and 0.3% (ST) (W/W). The electrical conductivity (EC) of the CK and ST treatments were 1.06 and 5.69 mS cm^−1^, respectively. The designed salt concentration (from 0 to 0.3%) was reached by increasing 0.1% every 24 h to avoid abrupt osmotic shock. Each group comprised 3 replicates with 12 seedlings in each replica. The cultivation was performed for 60 days.

### 2.2. Salt Injury and Growth Parameters

Leaf injury from salt stress was recorded on the 5th, 10th, 15th, 20th, 30th, 45th, and 60th days. A regime with 7 levels was defined as different salt damage levels (SDLs) according to a percentage of the area of damaged leaves: L0, no injury; L1, 5–10% injury; L2, 10–20% injury; L3, 20–40% injury; L4, 40–60% injury; L5, 60–90% injury, and few dead seedlings; L6, >90% injury and half the seedlings died; L7, all seedlings died.

Meanwhile, 2–3 leaves in each replicate were sampled and weighed immediately as the fresh weight (FW). Then, the leaves were immersed in distilled water for 24 h at 4 °C in the dark and weighed to obtain turgid weight (TW) after removing superficial moisture. Subsequently, the samples were oven-dried at 60 °C for 72 h and measured as the dry weight (DW). The leaf relative water content (%) was calculated as LRWC = 100 × [(FW − DW)/(TW − DW)] [25].

At the end of cultivation, five entire plants in each replica were harvested for biomass evaluation. Roots, stems, and leaves were weighed separately after washing and dried in an oven at 105 °C to constant weight (as dry weight) to calculate the root–shoot ratio.

### 2.3. Staining of ROS

ROS accumulation (superoxide anion and H_2_O_2_) was monitored by staining with Nitrotetrazolium blue chloride (NBT, CAS: 298-83-9) and 3,3′-Diaminobenzidine (DAB, CAS: 7411-49-6), respectively [26]. According to the previous work [22], 3 seedlings (CK and ST, respectively) leaves were selected on the 5th and 30th days for histochemical localization of ROS. Intact leaves were infiltrated with NBT (1 mg/mL) and DAB (1 mg/mL) for 2–6 h in darkness, and the reaction was stopped by dipping leaves into distilled water. Afterward, the pigments were removed with 75% ethanol and 5% glycerin in a boiling water bath. Finally, the cleared leaves were photographed. H_2_O_2_ was visualized as a reddish-brown color, while superoxide radicals were detected as blue color formazan.

### 2.4. Physiological Indicators

The upper-middle leaves of 5 seedlings were randomly collected on the days of 0th, 5th, 10th, 15th, 20th, 30th, 45th, and 60th for the following measurements.

#### 2.4.1. Electrolyte Leakage

The relative electrical conductivity (REC) was measured using a DJS-1D conductivity meter (Leica, Shanghai, China). Equal quantities (0.1 g) of mixed samples were cut into roughly sized squares and immersed in 20 mL of deionized water and shaken for 2 h, and the initial electrical conductivity (C_1_) of the solution was recorded. The samples were then boiled for 20 min and cooled to room temperature, and the final electrical conductivity (C_2_) was measured. The percentage of electrolyte leakage was calculated according to the formula REC(%) = (C_1_/C_2_) × 100 [27].

#### 2.4.2. Content of malondialdehyde (MDA) and the Antioxidant Enzyme Activities

Malondialdehyde (MDA), as the indicator of lipid peroxidation, was estimated using thiobarbituric acid (TBA) [28]. We measured the activities of superoxide dismutase SOD and catalase CAT with a total superoxide dismutase assay kit (Solarbio, Beijing, China) and catalase assay kit (Visible light) (Nanjing Jiancheng Bioengineering Institute, Nanjing, China), respectively.

#### 2.4.3. Osmotic Regulators’ Assays

As the main osmotic regulators, the contents of proline (Pro), soluble sugar (SS), and soluble protein (SP) were determined by the sulfosalicylic acid, anthrone colorimetry, and coomassie brilliant blue G-250 method [29], respectively.

#### 2.4.4. Photosynthetic Pigment Content Assays

The contents of chlorophyll a (Chl a), chlorophyll b (Chl b), and carotenoids (Caro) were determined according to the methods described by Wang and Huang [30]. Pigments were extracted from 0.2 g of fresh leaves using 12 mL of 95% ethanol (*v*/*v*) and filled to 25 mL using the same solvent. The absorbance was measured at 470, 649, and 665 nm (A_470_, A_649_, and A_665_, respectively), and the concentration and content of each pigment were calculated according to the formula described by Wang and Huang [30].

### 2.5. Photosynthetic Parameters

#### 2.5.1. Leaf Gas Exchange Measurements

Leaf gas exchange parameters, including light-saturated net photosynthetic rate (P_n_), stomatal conductance (gs), transpiration rate (T_r_), internal carbon dioxide concentration (C_i_), and leaf water use efficiency (WUE), were measured between 9:00 a.m. and 11:00 a.m. on 3 individuals in each replica using CIRAS-3 portable photosynthesis system (PP Systems Inc., Amesbury, MA, USA). Measurements were carried out under saturating photosynthetically active radiation (PAR) of 1200 μmol·m^−2^·s^−1^, a leaf chamber temperature of 25 ± 1 °C, relative humidity of 90%, and an atmospheric CO_2_ concentration of 450 ± 10 μmol·CO_2_ mol^−1^. Meanwhile, net photosynthetic rates were determined at gradient levels of PAR (0, 100, 200, 400, 600, 800, 1000, 1200, 1400 μmol·m^−2^·s^−1^). Values of P_n_ were plotted against PAR and fitted to P_n_–PAR response curves with Photosynthetic calculation 4.1.1 software using a nonrectangular hyperbola model as explained by Ye [31]. Based on this function, the maximum net photosynthetic rate (P_max_) was calculated.

#### 2.5.2. Photosystem-II (PSII) Efficiency Measurements

Chlorophyll fluorescence parameters were recorded using a portable Handy PEA (Plant Efficiency Analyzer-2126) fluorometer (Hansatech Instruments Ltd., Kings Lynn Norfolk, UK) on the same seedlings for photosynthetic leaf gas exchange measurements. The maximum fluorescence (F_m_) and minimum fluorescence (F_0_) were recorded on the third leaf using leaf clips after adaptation to darkness for 20 min. Then, variable fluorescence (F_v_) (calculated as F_m_ − F_0_), the maximal photochemical efficiency of photosystem II (F_v_/F_m_), and the potential photochemical activities of PSII (F_v_/F_0_) were calculated according to Baker and Rosenqvis [32].

### 2.6. RNA Extraction and qRT-PCR

Leaves for RNA extraction were collected from seedlings on the 5th and 30th days of cultivation. Total RNA was extracted using the RNA prep Pure Plant Kit (Qiagen, Beijing, China). After a visual check of RNA integrity on agarose gels and quantification using a NanoDrop 2000C spectrophotometer (Thermo Fisher Scientific, Waltham, MA, USA), total RNA was reverse transcribed using the PrimeScript™ II 1st Strand cDNA Synthesis Kit (Takara Bio Inc., Otsu, Japan). Quantitative real-time PCR (qRT-PCR; StepOnePlus™ Real-Time PCR System, Thermo Fisher Scientific, Waltham, MA, USA) was used to detect the expression pattern of genes related to photosynthesis and photorespiration, which were explored from transcriptome data of *C. hongkongensis* subsp. *tonkinensis* (data unpublished). The primers shown in Appendix A were designed using NCBI and Primer Premier 5.0 software [33]. PCR reaction (20 µL) contained 10 µL of SYBR Green Real-time PCR Master Mix (TOYOBO Co., LTD., Osaka, Japan), 0.2–0.4 μg of template cDNA, and 0.5 µM of each primer. The PCR procedure was set at 94 °C for 40 s, 40 cycles of 94 °C for 10 s, 55 °C for 30 s, and 72 °C for 35 s. PCR products were verified by melting curve analysis of a single peak using agarose gel electrophoresis (Appendix A). All reactions were performed with biological triplicates as well as three technical replicates. The relative expression of genes at the transcription level was calculated using the 2^−∆∆CT^ method [34]. Gene expression was normalized to the stably expressed glyceraldehyde-3-phosphate dehydrogenase gene (GAPDH) [35].

### 2.7. Data Analysis

All data were subjected to Duncan’s new multiple ranges (DMR) tests, independent sample *t*-tests, and one-way analyses of variance (ANOVAs). Pearson’s correlation analysis was performed to determine the relationship between the response variables using SPSS 19.0 software (IBM Corporation, Armonk, NY, USA). Differences at *p* < 0.05 were considered significant. All parameters are represented as the means ± SD (*n* = 3).

## 3. Results

### 3.1. Symptoms of Salt Damage

The degree of salt injury was exacerbated as the duration expanded (Table 1). As the salt damage level (SDL) increased, symptoms of salt damage expanded from the bottom to the tip of the leaves. No distinct damage symptoms appeared on the seedlings during the first five days of salt stress (Figure 1A). Obviously, on the 30th day, symptoms of salt injury, such as yellowing, scorching, and withering around the leaf margin (L4) (Figure 1A), were observed in 45.5% of seedlings, and a few dead individuals (8.3%) were found as well. On the 60th day, the mortality rate reached 41.7%, and the survival seedlings were leafed scarcely. Compared with CK, the roots of survival individuals subjected to salt stress were dark and necrotic, and a small number of new roots emerged (Figure 1B).

### 3.2. Changes in Leaf Relative Water Content (LRWC) and Biomass Allocation

As shown in Figure 2, no significant variations in LRWC were noticed before 30 days of salt stress, corresponding to slight stress-induced symptoms (Figure 1). Notably, a steep decrease in LRWC in the salt-stressed seedlings occurred after 30 days when compared to that of CK (*p* < 0.05; Figure 2); the values in stressed seedlings were reduced by 9.85% and 28.17% on the 30th and 60th days, respectively.

Generally, adjustment of the biomass distribution in the roots, stems, and leaves of plants is an effective way to adapt to adverse conditions. In comparison, the biomass of various organs (root and leaf) and whole plants treated with 0.3% salt were significantly lower than that of CK after 60 days of cultivation (*p* < 0.01; Table 2). However, no significant difference in the underground/aboveground biomass ratio occurred between CK and salt-treated seedlings (Table 2).

### 3.3. Changes in Levels of ROS, Cell Membrane Damage, and Response to Antioxidant Enzymes

On the 5th day of cultivation, a small amount of superoxide anion (NBT staining) was accumulated in stressed leaves instead of unstressed leaves (Figure 3). No signal of H_2_O_2_ (DAB staining) appeared in the leaves of CK and ST (Figure 3). At the synchronous physiological level, a significant increase in REC and MDA concentrations occurred in leaves of ST (*p* < 0.05), indicating oxidative stress in leaves of five-days stress (Figure 4a,b). Compared with CK, a significant decrease in SOD activity (*p* < 0.05; Figure 4c), and a slight increase in CAT (*p* > 0.05; Figure 4d) were measured in leaves of ST. Prolonged to the 10th stress, REC and MDA concentrations in leaves of ST fell to levels of CK.

After 30 days of salt stress, increasing aggregation of superoxide anion could be seen, but only a small amount of H_2_O_2_ accumulated in leaves of ST until the 60th day (Figure 3). Meanwhile, both MDA and REC contents in leaves of ST increased obviously, reaching 43.89% and 48.07% above CK (Figure 4a,b). As an extension of salt-stressed, both values in ST rose sharply, jumping to the maximum of 0.05 μmol g^−1^ and 67.37%, respectively; however, the two parameters fluctuated slightly in CK (Figure 4a,b). Inconsistently, both SOD and CAT activity in stressed seedlings were higher than that of CK, but at an insignificant level (*p* > 0.05) (Figure 4c,d).

### 3.4. Production of Osmotic Regulators

Although the Pro contents in stressed seedlings were always higher than those in CK, no significant difference was noticed between them over time except that on the 10th and 60th days (Table 3). As the stress continued, the SS and SP contents displayed upward trends, and significant differences were detected between CK and ST after 15 days (*p* < 0.05). On the 30th day of stress, the contents of SS and SP increased by 80.66% (*p* < 0.01) and 11.92% (*p* < 0.05), respectively (Table 3). In addition, all tested parameters reached their maximum on the 60th day. These results indicate that maintaining osmotic balance in stressed seedlings of *C. hongkongensis* subsp. *tonkinensis* were realized mainly by enhancing the contents of osmotic regulation substances. Among them, SS could play a more critical role than the other two substances.

### 3.5. Photosynthetic Performance

Extending the time of salt-stress duration had a pronounced effect on the rate of net photosynthesis (P_n_) (Figure 5b). With the extension of stress duration, the maximum net photosynthetic rate (P_max_) gradually declined, and the maximum reduction was 84% in ST in comparison to CK at the end of treatment (CK: 8.757 µmol·(CO_2_)·m^−2^·s^−1^; ST: 1.428 µmol·(CO_2_)·m^−2^·s^−1^; Figure 5a). Likewise, significant variations were recorded for the gas exchange characteristics of seedlings under salinity conditions. On the 5th day of stress, P_n_ in ST (2.57 µmol (CO_2_)·m^−2^·s^−1^) significantly declined by 64.68% in comparison with CK (7.27 µmol (CO_2_)·m^−2^·s^−1^) (*p* < 0.01; Figure 5b); similar significant reductions of 37.64%, 56.94%, and 81.72% were observed in C_i_, T_r_, and gs of treated seedlings in comparison to CK, respectively (*p* < 0.01; Figure 5c,e), but no significant changes in WUE occurred (Figure 5f). On the 30th day of stress, the fluctuation of C_i_ was inversely proportional to that of P_n_, with a gradual increase of 20.38% in C_i_, and a corresponding significant decrease of 86.73% in P_n_ in comparison to CK (Figure 5b,c). At this time, there was also a decrease in gs, T_r_, and WUE of 70.61%, 59.70%, and 66.93% in stressed seedlings compared with CK, respectively (*p* < 0.01; Figure 5d–f). After 30 days, P_n_ and C_i_ in ST maintained a relatively stable state except for a large fluctuation in WUE.

The maximum quantum yield of PSII (F_v_/F_m_) and potential activities of PSII (F_v_/F_0_) changed slightly before 30 days of stress, then began to drop abruptly, and finally down to the minimum on the 60th day (*p* < 0.01; Figure 5g–h). Both variables were 28.37% and 61.54% of CK at the end of treatment, respectively.

Table 4 shows no significant difference in the chlorophyll a, chlorophyll b, total chlorophyll, and carotenoid contents between CK and ST before 45-day stress (*p* > 0.05). Subsequently, all parameters of photosynthetic pigments gradually declined. Notably, a significant drop (*p* = 0.011) in chlorophyll a/b (Chl a/b), reflecting the photosynthetic activity of the leaves, was observed on the 60th day, indicating that the photosynthetic activity of leaves was weakened after a longer duration of salt stress.

### 3.6. Transcriptional Expression of Genes in the Pathways of Photosynthesis and Photorespiration

Photosynthetic parameters, such as P_n_, C_i_, T_r_, and gs, responded to salt stress quickly on the 5th day of stress (Figure 5), suggesting that photosynthesis response to salt stress happened at this time. To better understand the mechanism of the photosynthetic response at the transcriptional level, differentially expressed genes related to key photosystem II, photosynthetic pigment, and photorespiration genes at the different stressed stages were enriched (Appendix A) [13]. Remarkably, differential genes on the 5th and 30th days of salt stress were emphatically analyzed, when significant drops in P_n_ and changes with V-type in C_i_ occurred.

On the 5th day of salt stress, in the photosystem II pathway, the transcription level of gene encoding protein D1 (chloroplast), namely, *psbA*, was upregulated significantly. In contrast, the transcription level of the gene encoding the PSB28 protein, namely, *psb28*, decreased, and that of the *LFNR* gene coding for ferredoxin-NADP^+^ oxidoreductase remained stable. Simultaneously, regarding genes responsible for photosynthetic pigment synthesis, the stress reactions at the transcriptional level of chlorophyll synthase *CHLG* and phytoene desaturase *PDS* were not triggered by salt stress, which was consistent with the contents of chlorophyll and carotenoids at that time (Figure 6; Table 4). Significantly rising transcript accumulations of *SGAT* and *GGAT* in photorespiration pathways, as well as *PGK* coding for phosphoglycerate kinase in the Calvin cycle, were detected in the stressed leaves. However, the transcripts of the *GLYK* gene coding for D-glycerate 3-kinase, the *Fd-GOGAT* gene coding for ferredoxin-dependent glutamate synthase in photorespiration, and the *CYFBP* gene coding for fructose-1,6-bisphosphatase in the Calvin cycle presented no obvious stress response.

Another key point for photosynthetic reaction occurred on the 30th day of stress. The expression levels of three key genes involved in photosynthesis, namely, *psbA*, *psb28*, and *LFNR*, were downregulated significantly, which was following the significantly lowered values of F_v_/F_m_ and F_v_/F_0_ at that time (Figure 5g,h). However, a corresponding reduction occurred only on the PDS gene rather than the CHLG gene. The transcription level of *GGAT* and *GLYK* on the photorespiration pathway showed a significant decrease, as did the transcription level of *PGK* in the Calvin cycle (Figure 6), whereas the transcription levels of *SGAT*, *FdGOGAT*, and *CYFBP*, which are responsible for photorespiration and the Calvin cycle, respectively, changed insignificantly in salt-stressed seedlings.

## 4. Discussion

### 4.1. Prolonged Salt Stress Affects the Survival Rate and Phenotypic Traits

The survival rate of *C. hongkongensis* subsp. *tonkinensis* seedlings suffering from 0.3% salt stress decreased with prolonged duration, and a half survival rate occurred on the 60th day in this study. The tolerance of single salt stress is stronger than that of integrated stress from salt and heat in this species, whose survival was only 50% on the 20th day [7]. As an adjunct to assess the salt tolerance of seedlings [36], salt injury symptoms of leaves in our study were aggravated along with rising mortality, which was more serious than that of red-osier dogwood (*Cornus sericea* L.), a moderate-tolerant species [37]. Although salt injury in *C. hongkongensis* subsp. *tonkinensis* developed slowly with a high LRWC before 30 days of stress, the rising mortality implies that this species may not endure 0.3% salt stress for a long period. However, frequent rainfall in coastal regions of the subtropical area lowers salt content in the soil, which can alleviate salt stress temporarily. Thus, *C. hongkongensis* subsp. *tonkinensis* could survive well when rainfall occurs constantly, which was supported by a 5-year-old plantation with good performance on the Dafeng forest farm.

Alteration of biomass allocation is a comprehensive response to salt stress, reflecting not only in quantitative changes but also in redistribution patterns among various organs [38]. As reductions of biomass happened in most plants suffering from salt stress, such as kiwifruit, *C. hongkongensis* subsp. *elegans*, and *C. florida* [19,22,24], a similar reduction in *C. hongkongensis* subsp. *tonkinensis* is consistent with the decrease in P_n_ (r = 0.815, *p* < 0.05). However, no significant redistribution of biomass in this species under salt stress differs from *C. sericea* and *Olea europaea* L. [37,39]. Presumably, the reduction in biomass is too much to rearrange among various organs.

### 4.2. Salt Stress Response of Oxidative Stress and Osmotic Defense

Reactive oxygen species (ROS) include superoxide free radical ($O_{2}^{-}$), hydrogen peroxide (H_2_O_2_), singlet oxygen (^1^O_2_), hydroxyl free radical (·OH), etc., and the accumulation of ROS can cause peroxide stress in plants [40]. Upon five days of saline stress, an accumulation of $O_{2}^{-}$ with a significant increase in the content of MDA, derived from ROS-damaged membranes, was observed in salt-treated leaves instead of CK, inferring oxidative damage in the membrane in our study [41]. As Docimo et al. [42] suggested that a significant decline of SOD activity in *Cynara cardunculus* L. could indicate minor damage to the enzymatic machinery under short saline stress, it also happens in *C. hongkongensis* subsp. *tonkinensis*. Subsequently, the drop of MDA to levels of CK versus unchanged SOD and CAT enzyme activities suggest that the latter two enzymes play small roles in scavenging ROS under short-term stress. After 30 days of salt stress, integrating the accumulation of $O_{2}^{-}$, an increase of MDA and REC in leaves of ST hint at a disrupted cell membrane, which is similar to the salt response in *Psidium guajava* L. [4].

In general, electrolyte leakage resulting from damage to cell membrane under salt stress could trigger changes in soluble osmotic regulating substances’ (i.e., proline, soluble sugars, and soluble proteins) accumulation [19,43]. In our study, the content of SS, which was positively correlated with REC (r = 0.731, *p* < 0.01) and negatively correlated with LRWC (r = −0.817, *p* < 0.01), significantly increased with prolonged salt stress (Appendix A). It infers that SS could act as the first line of defense in *C. hongkongensis* subsp. *tonkinensis* to maintain cellular moisture for the long term [19]. In addition, the significant increase in SP, an important index to describe the degree of cell membrane damage, indicates that the cell membrane had been damaged after 30 days of stress [44]. This is also supported by the high MDA and REC in ST. We speculate that the osmoregulation mode in *C. hongkongensis* subsp. *tonkinensis* under salt stress starts from SS accumulation, followed by SP, and Pro in turn.

### 4.3. Responses of Stomatal and Nonstomatal Components of Photosynthesis to Salinity

In general, the first physiological response to salt stress is to avoid water loss through transpiration, which is achieved via stomatal closure leading to a decline in gs values [25]. Under short saline stress (5 d), notable reductions in C_i_ and P_n_ parallel to gs decline suggest that it could be attributed to stomatal closure. It is consistent with that of *C. hongkongensis* subsp. *elegans* [7]. Significantly, changes in measurements (Ci, Tr, gs, and WUE) also happened in CK, and it is probably caused by fluctuated temp in the culture room, which is a half-opening environment. Despite the significant decline in gs and T_r_ within 30 days of stress, WUE in *C. hongkongensis* subsp. *tonkinensis* remained stable to adapt to osmotic/water deficits, in accordance with stable LRWC [45]. This was in agreement with the WUE response in red-osier dogwood under similar salt concentrations [37]. Additionally, higher expression of *CHLG* in the transcripts, as well as constant chlorophyll content, suggest that stability in chlorophyll content may be another physiological adaptation mechanism maintaining photosynthesis [46,47]. A similar response was also presented in *C. florida* L. and *C. kousa* F. Buerger ex Hance [22].

Notably, both significances have detected the increase in C_i_ vs. decreases in P_n_, P_max_, T_r_, and gs with prolonged saline stress (30 d). A similar result was found in salt-stressed melon as well [6]. Such alterations in photosynthetic indicators under high salt concentrations may be caused by nonstomatal restriction factors [48,49]. Concurrently, the significant decreases in F_v_/F_m_ and F_v_/F_0_, which were the indicators of PSII reaction center activity in leaves, were significantly correlated with these photosynthetic parameters (e.g., P_n_, T_r_, and gs) (Appendix A). It ulteriorly proved that the drop in P_n_ was mainly due to the damage of PSII after 30 d stress. The final decrease in chlorophyll content and WUE also account for the reduction in photosynthesis in *C. hongkongensis* subsp. *tonkinensis* under long-term salt stress [47,50]. Alternatively, the negative correlation between photosynthesis and leaf injury (r = −0.899, *p* < 0.01) suggests that the reduction of photosynthetic rate can explain the reason for the degree of leaf injury (Appendix A).

### 4.4. Protective Effect of Photorespiration on Photosynthetic System

In linear electron flow (LEF), electrons are transferred from ferredoxin to NADPH by the photosynthetic FNR enzyme; then, NADPH and ATP are used for carbon fixation in the Calvin cycle [10]. On the 5th day of salt stress, the slowly upregulated expression of LFNR was significantly correlated with the expression of enzymes in the Calvin cycle (e.g., CYFBP and PGK). The expression of LFNR was also positively correlated with F_v_/F_m_ and F_v_/F_0_, but the correlation coefficients were lower than that of enzymes in the Calvin cycle. We hypothesize that short-term salt stress may promote a better carbon fixation ability. However, more direct evidence should be provided to further prove this deduction. It is a hot spot where photorespiration can protect a photosynthetic system against oxidative damage under salt stress [13]. Photorespiration contributes to protecting PSII from oxidative stress by accelerating the repair rate of the D1 protein, encoded by *psbA* [9]. In our study, a negative impact on chlorophyll fluorescence was not observed before 30 days of salt stress, suggesting an uninhibited PSII in the short term [6]. Lu et al. [7] also found a similar result in *C. hongkongensis* subsp. *elegans*. Interestingly, significant upregulation in *psbA* transcripts and stability of F_v_/F_m_, F_v_/F_0_ on the 5th day, sharp downregulation in *psbA* transcripts and decrease in F_v_/F_m_, F_v_/F_0_ on the 30th day (r = 0.602, *p* < 0.01; r = 0.598, *p* < 0.01, respectively; Appendix A) imply that *psbA* may play a role in protecting PSII from photoinhibition. Moreover, for a constant decline in F_v_/F_m_ and F_v_/F_0_, the downregulated expression of both *LFNR* and *psbA* indicated that prolonged salt stress resulted in PSII photoinhibition (Appendix A) [8,51], and it appeared in *C. florida* as well [24].

The upregulated expression of two transaminases (e.g., SGAT and GGAT) could accelerate the operation of the photorespiration pathway and control the cell damage caused by salt stress [12]. Yang et al. [14] stated that the improvement of photorespiration stimulated the antioxidant system to reduce the accumulation of ROS. However, on the 5th day of salt stress, an increase in O^2−^, the transcriptional levels of *SGAT* and *GGAT*, as well as a decline in SOD activity, demonstrate that photorespiration did not strengthen the antioxidant enzyme system in *C. hongkongensis* subsp. *tonkinensis*. Notably, the increase in transcription levels of *GGAT* and *SGAT* may ensure the stability of PSII and the water-retaining capability of leaves; this is proved by the significant positive correlations of *GGAT* vs. F_v_/F_0_ (r = 0.483, *p* < 0.05) and *SGAT* vs. WUE (r = 0.578, *p* < 0.05) (Appendix A). These assumptions are also proved by the results in wild-type duckweed (*Lemna mino*) and *Solanum lycopersicum* L. [14,15]. On the 30th day of salt stress, slower upregulated expression of *SGAT* and significantly downregulated expressions of photorespiration-metabolizing enzyme genes, such as *GGAT*, *GLYK*, and *FdGOGAT*, indicate depressed photorespiration [12]. Concurrently, the downregulated expressions of *PGK* as well as inhibited PSII indicate that salt stress may alter photosynthesis by altering the expression of related enzymes in the Calvin cycle [52]. It is also proved by their correlation of PGK vs. F_v_/F_m_ (r = 0.493, *p* < 0.05) and PGK vs. F_v_/F_0_ (r = 0.513, *p* < 0.05). Overall, photorespiration may play a critical role in safeguarding the photosystem under short-term stress.

## 5. Conclusions

*C. hongkongensis* subsp. *tonkinensis* is a potential ornamental tree species for afforest in areas affected by moderate salt up to 0.3%. P_n_ is very sensitive to salt stress; both stomatal and nonstomatal limitations play important roles in reducing the photosynthesis rate in this species. In the early stage of salt stress (5 d), stable WUE and chlorophyll content were the positive physiological responses regulating osmotic balance and maintaining normal photosynthetic activity. Meanwhile, the significantly upregulated expression of *psbA*, *SGAT*, and *GGAT* under salt stress could protect the photosystem. In the late period (30 d), osmotic adjustment substances, mainly soluble sugars, played a positive role in regulating the osmotic balance. Prospectively, the function of enzyme genes related to photorespiration should be further studied to better understand the salt tolerance mechanism at the level of photosynthesis in this species.

## Figures and Tables

**Figure 1 plants-11-00940-f001:**
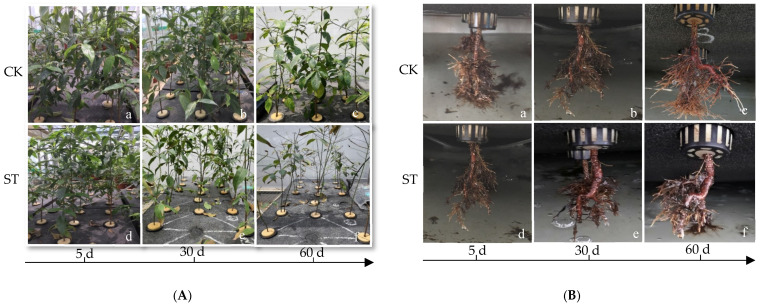
Leaves and roots growth performances of *C. hongkongensis* subsp. *tonkinensis* seedlings grew in the nutrient solution without (CK)/with (ST) 0.3% salt on the days of 5th, 30th, and 60th. (**A**,**B**) represent leaves and roots; (**a**–**c**) represents seedings of CK, (**d**–**f**) represents seedings of salt-treated (0.3% salt), respectively.

**Figure 2 plants-11-00940-f002:**
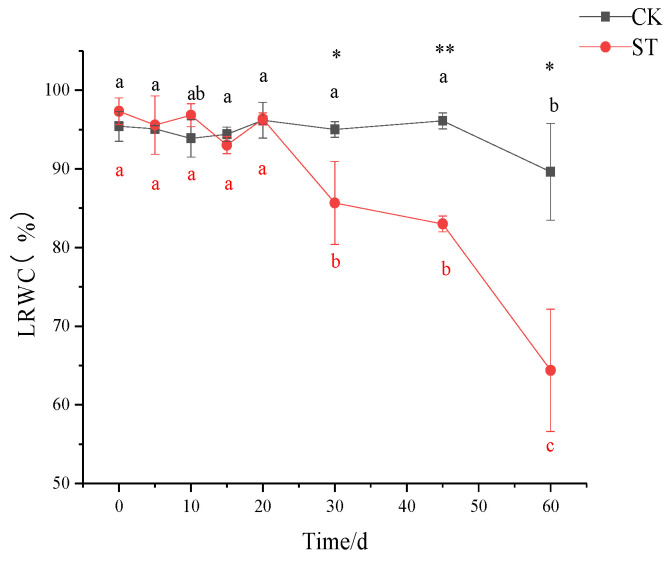
Response of leaf relative water content (LRWC) in the seedlings of *C. hongkongensis* subsp. *tonkinensis* to salt stress. * and ** indicate a significant difference between CK and ST at the same time according to the independent-samples *t*-test (*p* < 0.05, 0.01); different lowercase letters indicate significant differences among tested time within treatment (*p* < 0.05) according to one-way analyses of variance (ANOVAs). The same is below.

**Figure 3 plants-11-00940-f003:**
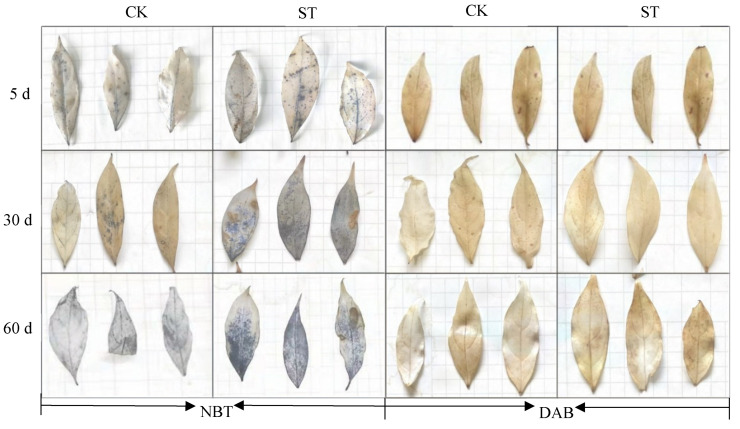
Changes in ROS accumulation in leaves of *C. hongkongensis* subsp. *tonkinensis* under salt stress.

**Figure 4 plants-11-00940-f004:**
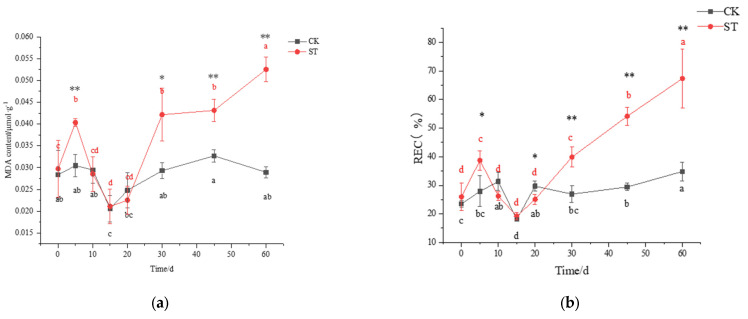
Changing pattern of membrane damage and antioxidant enzyme activity in leaves of *C. hongkongensis* subsp. *tonkinensis* under salt stress. (**a**) represents the content of malondialdehyde (MDA), (**b**) represents the content of relative electrical conductivity (REC), (**c**) represents superoxide dismutase activity (SOD), (**d**) represents catalase activity (CAT). * and ** indicate a significant difference between CK and ST at the same time according to the independent-samples *t*-test (*p* < 0.05, 0.01); different lowercase letters indicate significant defenses among tested time within treatment (*p* < 0.05) according to one-way analyses of variance (ANOVAs). The same is below.

**Figure 5 plants-11-00940-f005:**
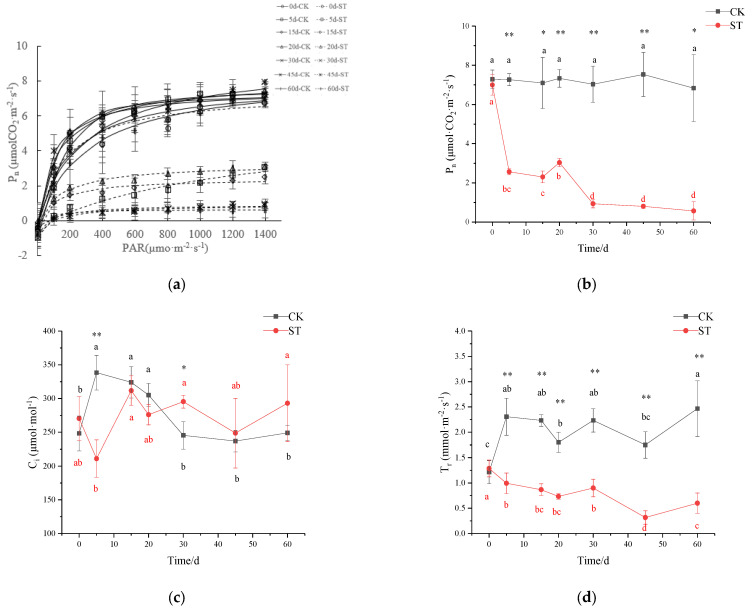
Effect of salinity stress on plant photosynthetic performance of *C. hongkongensis* subsp. *tonkinensis*. * and ** indicate a significant difference between CK and ST at the same time (*p* < 0.05, 0.01); different lowercase letters indicate a significant difference among the tested time within treatment (*p* < 0.05). Black is CK, red is ST. Data missing on the 10th day. (**a**) represents P_n_–PAR response curves; (**b**–**f**) represent photosynthetic rate (P_n_), internal carbon dioxide concentration (C_i_), transpiration rate (T_r_), stomatal conductance (gs), and leaf water use efficiency (WUE), respectively; (**g**,**h**) represent the maximal photochemical efficiency of photosystem II (F_v_/F_m_) and potential activities of PSII (F_v_/F_0_).

**Figure 6 plants-11-00940-f006:**
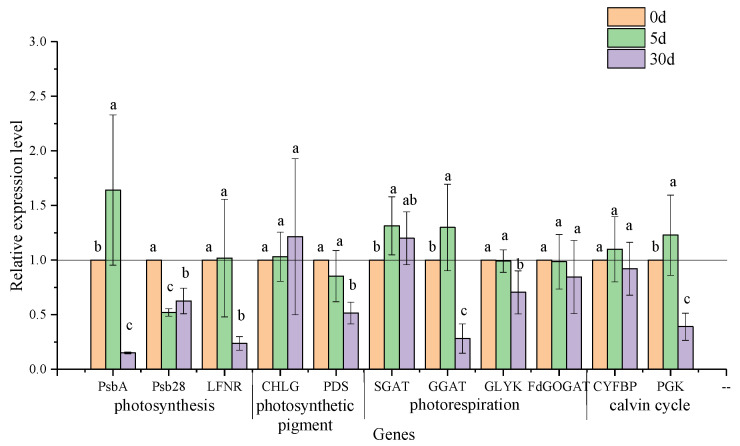
Effects of salt stress on the relative expression of genes involving photosynthesis and photorespiration in leaves of *C. hongkongensis* subsp. *tonkinensis*. Each value represents the mean ± SD of three biological and three technical replicates (The value of all genes on the 0the day defaults to 1). Values marked with different lowercase letters indicate a significant difference between sample times at *p* < 0.05.

**Table 1 plants-11-00940-t001:** Grade of salt damage changed with elongation of salt stress in seedlings of *C. hongkongensis* subsp. *tonkinensis*.

Duration of Salt Stress/d	Level of Salt Damage	Salt Damage Rate/%	Mortality Rate/%
0	L0	0.0	0
5	L0	4.2	0
10	L1	8.3	0
15	L2	12.5	0
20	L3	25.0	0
30	L4	45.5	8.3
45	L5	89.5	20.8
60	L6	100.0	41.7

**Table 2 plants-11-00940-t002:** Response of biomass allocation in the seedlings of *C. hongkongensis* subsp. *tonkinensis* to salt-stress for 60 days.

Biomass/g	Treatment		Sig. of Difference
	CK	ST	
Root	1.86 ± 0.20	1.25 ± 0.27	*
Stem	3.96 ± 0.58	3.41 ± 0.67	NS
Leaf	1.94 ± 0.52	0.32 ± 0.14	**
Total	7.76 ± 1.29	4.98 ± 1.06	*
Underground/aboveground biomass ratio	0.33 ± 0.03	0.36 ± 0.02	NS

Notes: * *p* < 0.05, ** *p* < 0.01; NS indicates no significant difference between CK and ST according to an independent-samples *t*-test.

**Table 3 plants-11-00940-t003:** Effects of salt stress on osmotic regulators in the seedlings of *C. hongkongensis* subsp. *tonkinensis*.

Osmotic Regulators	Treatments	Time/d							
		0	5	10	15	20	30	45	60
PRO (μg·g FW^−1^)	CK	596.41 ± 32.80 a	551.74 ± 34.39 ab	547.39 ± 13.15 ab	576.62 ± 20.90 ab	558.19 ± 9.12 ab	589.78 ± 52.59 a	576.62 ± 20.90 ab	534.49 ± 9.12 b
	ST	547.66 ± 33.13 c	575.98 ± 26.33 c	630.17 ± 23.23 bc	592.42 ± 41.80 c	568.72 ± 33.51 c	592.42 ± 59.64 c	691.15 ± 83.78 b	911.01 ± 25.39 a
	*t*-test			**					**
SS (mg·g FW^−1^)	CK	4.23 ± 0.86 a	3.89 ± 0.79 ab	2.87 ± 0.79 ab	3.87 ± 0.98 ab	2.47 ± 0.42 b	3.00 ± 0.30 ab	3.13 ± 1.05 ab	3.82 ± 0.13 ab
	ST	3.84 ± 1.41 c	4.05 ± 0.82 c	3.56 ± 1.15 c	5.03 ± 0.94 bc	4.33 ± 0.64 c	5.42 ± 0.18 bc	6.35 ± 0.49 b	9.87 ± 1.85 a
	*t*-test					*	**	**	**
SP (mg·g FW^−1^)	CK	3.44 ± 0.12 a	3.17 ± 0.23 b	3.38 ± 0.10 ab	3.42 ± 0.08 ab	3.38 ± 0.11 ab	3.54 ± 0.04 a	3.59 ± 0.11 a	3.43 ± 0.20 a
	ST	3.24 ± 0.47 b	3.13 ± 0.12 b	3.32 ± 0.25 b	3.49 ± 0.11 b	3.50 ± 0.28 b	3.96 ± 0.24 a	4.10 ± 0.11 a	4.40 ± 0.19 a
	*t*-test						*	**	**

Note: * and ** indicate a significant difference between CK and ST at the same time (*p* < 0.05, 0.01); different lowercase letters indicate a significant difference among tested time within treatment (*p* < 0.05).

**Table 4 plants-11-00940-t004:** Modulations in chlorophyll and carotenoid contents in *C. hongkongensis* subsp. *tonkinensis* under salt stress.

Photosynthetic Pigment Content	Treatments	Duration/d							
		0	5	10	15	20	30	45	60
Chla (mg g^−1^)	CK	0.86 ± 0.02 e	0.87 ± 0.10 e	1.13 ± 0.11 c	1.11 ± 0.08 cd	0.94 ± 0.05 de	1.02 ± 0.18 cde	1.30 ± 0.07 b	1.55 ± 0.06 a
	ST	0.89 ± 0.09 ab	0.91 ± 0.12 ab	1.06 ± 0.06 a	1.00 ± 0.04 ab	0.90 ± 0.08 ab	0.77 ± 0.15 b	0.84 ± 0.05 ab	1.07 ± 0.24 a
	*t*-test							**	*
Chlb (mg g^−1^)	CK	0.27 ± 0.04 e	0.33 ± 0.04de	0.43 ± 0.02 c	0.43 ± 0.04 c	0.36 ± 0.02 cd	0.39 ± 0.07 cd	0.52 ± 0.03 b	0.60 ± 0.03 a
	ST	0.28 ± 0.02 c	0.36 ± 0.04 bc	0.43 ± 0.02 ab	0.40 ± 0.02 b	0.36 ± 0.03 bc	0.31 ± 0.06 c	0.36 ± 0.00 bc	0.49 ± 0.09 a
	*t*-test							**	
Total Chl (mg g^−1^)	CK	1.12 ± 0.06 f	1.21 ± 0.14 ef	1.57 ± 0.12 c	1.54±0.12 cd	1.30 ± 0.07 def	1.42 ± 0.24 cde	1.82 ± 0.10 b	2.15 ± 0.10 a
	ST	1.17 ± 0.08 bc	1.27 ± 0.17 abc	1.49 ± 0.08 ab	1.40 ± 0.06 ab	1.26 ± 0.11 abc	1.08 ± 0.20 c	1.19 ± 0.05 bc	1.56 ± 0.33 a
	*t*-test							**	*
Carotenoid (mg g^−1^)	CK	0.16 ± 0.04 c	0.17 ± 0.02 c	0.21 ± 0.03bc	0.21 ± 0.02 bc	0.18 ± 0.01 c	0.20 ± 0.03 bc	0.24 ± 0.02 b	0.29 ± 0.01 a
	ST	0.15 ± 0.03 ab	0.18 ± 0.02ab	0.19 ± 0.01ab	0.20 ± 0.02 ab	0.17 ± 0.01 ab	0.15 ± 0.02 c	0.17 ± 0.01 ab	0.20 ± 0.05 a
	*t*-test							**	
Chl a/b	CK	3.26 ± 0.42 a	2.64 ± 0.06 b	2.61 ± 0.17 b	2.58 ± 0.08 b	2.59 ± 0.05 b	2.61 ± 0.01 b	2.53 ± 0.08 b	2.60 ± 0.04 b
	ST	3.15 ± 0.49 a	2.52 ± 0.04 b	2.46 ± 0.07 b	2.49 ± 0.03 b	2.54 ± 0.02 b	2.50 ± 0.06 b	2.35 ± 0.15 b	2.18 ± 0.16 b
	*t*-test								*

Note: * and ** indicate a significant difference between CK and ST at the same time (*p* < 0.05, 0.01), different lowercase letters indicate a significant difference among tested time within treatment (*p* < 0.05).

## Data Availability

Data is contained within the article.

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
