# Peer review of "Responses of Physiology, Photosynthesis, and Related Genes to Saline Stress in *Cornus hongkongensis* subsp. *tonkinensis* (W. P. Fang) Q. Y. Xiang"

_plants, 2022, doi:10.3390/plants11070940_

Round 1

Reviewer 1 Report

In this manuscript, the authors analyzed the damage and various responses, mainly related to photosynthesis, of salt-tolerant Cornus honkonensis subsp. tokinensis seedlings under salt stress conditions. It then discusses how this plant adapts to salt stress.

In order to publish this manuscript, it is necessary to revise the following points.

Overall

  In this manuscript, the subspecies names are placed in the positions of the specific epithets in several plant scientific names, such as Corunus hongkogenesis sibsp. as C. tokinensis, which is an incorrect way to write scientific names.

  The authors have only compared the responses of the plant under salt stress and non-salt stress conditions. Comparison with a related plant that is not salt tolerant, such as the North American dogwood, should provide a better understanding of the salt tolerance of the plant of interest.

  The authors simply discuss whether the changes in the two measurements are correlated, but it is unlikely that a relationship with a low correlation coefficient and a relationship with a high correlation coefficient can be considered equally. It would be better to include a discussion regarding the magnitude of the correlation coefficient.

Materials and methods

Lines 108-110:  "The designed salt concentration was reached by increasing 0.1% every 24 h to avoid abrupt osmotic shock."

 It should be clearly indicated when the 0 day was. Also, is it correct to say that "salt" here means NaCl, of course?

Lines 113-117: Does the percentage of damage represent a percentage of the area of damaged leaves or a percentage of the number of damaged leaves? It should be clearly stated what percentage is being indicated.

Lines 128-135: In the measurement of ROS accumulation, the infiltration was performed under dark conditions with NBT or DAB for 2-6 hours. Since ROS is not produced very much in plants under dark conditions, ROS should have been scavenged by antioxidant enzymes and reduced during this time. Moreover, if the length of infiltration times was different, the amount of ROS scavenged would be different. Therefore, this observation does not seem to reflect the amount of ROS accumulated during growth and cannot be properly discussed, which may be the reason why H2O2 could not be detected by DAB staining.

Lines 140-143: In the electrolyte leakage measurement, the leaves were shredded, but is it not necessary to take into account the amount of electrolytes that was released from the destroyed cells?

Lines 147-156: It does not describe the conditions under which the plants were collected or the method of extraction.

Lines 154-156: In the osmotic regulator assay, there is no correspondence between the compounds measured and the assay described. Anthrone is a reagent used for sugar determination, while Comassie brilliant blue and sulfosalicylic acid are both considered to be used for protein determination. What method was used to quantify proline?

Line 169 Fluorescence seems to be a mistake for system.

Results

Figure 4 and 5: Even in CK, there are measurements that change very significantly from 0 day to 5 day (SOD, CAT, Ci, Tr, gs, and WUE). This may be due to a change in the growth environment at 0 day. If so, this should be clearly stated in the materials and methods. Also, if this point is not taken into consideration in the discussion, there is no point in comparing CK and ST.

Table 3: The cells in the leftmost column have line breaks.

Table 4: Why are only measurements related to chlorophyll shown in a table and not in a graph? Wouldn't a graph be better for comparison with other responses, especially those related to photosynthesis?

Line 337: The "D" in D-glycerate 3-kinase should be represented by a small capital.

Figure 6: Changes in gene expression over time under ST conditions are shown. In order to consider whether the expression of such genes contributes to salt stress tolerance, it may be necessary to compare the changes in expression with those under CK conditions to discuss.

Discussion

Lines 365-369: The relevance of this discussion to the results of this experiment is not clear. Do the authors believe that in a field with fluctuating salt concentrations, the period of salt stress is short enough for the target plants to tolerate it and grow well? It should be rewritten for easier understanding.

Lines 391-393: The authors attribute the increase in SOD activity under salt stress to an increase in Pro, which plays a protective role in enzyme activity. However, they have not confirmed the change in SOD expression under these conditions, and thus cannot discuss this far.

Lines 398-399: What is the relevance of this discussion to the results of this SP measurement in this manuscript?

Lines 415-420: It is mentioned that changes in Tr and gs may be caused by nonstomatal restriction factors, but what factors are assumed to be responsible? Can these two indicators be changed by factors other than stomatal opening/closing?

Lines 423-425: The reference No. 7 cited here describes the psbA gene in cyanobacteria. Cyanobacteria do not have an organelle, have a CO2-concentrating mechanism, and should not have high photorespiratory activity. In this paper, is there any mention of the relationship between D1 protein synthesis and photorespiratory activity?

Lines 432-433:  The reference No. 8 cited here only describes chlororespiration, which is a completely different physiological response than photorespiration. If the authors are writing this manuscript under the assumption that photorespiration and chlororespiration are the same metabolic system, they should rewrite the manuscript substantially.

FNR is an enzyme at the terminal of the linear electron flow of the photosystems and is also involved in the cyclic electron flow in chlororespiration. However, it is not a regulator of the electron transfer system; how can it be explained the link between the stable expression of FNR and the inactive state of electron transfer? And does not this contradict the statement in the last line that reduced expression of FNR leads to reduced PSI electron transport capacity? Also, since PSI electron transfer capacity was not measured in this study, it would not be possible to discuss about PSI.

Lines 442-445: Photorespiration suppresses ROS formation, not scavenges it.

Line 451: PGK is not a photorespiratory metabolizing enzyme, but an enzyme that functions in the Calvin cycle.

Line 462: What kind of change is described by reverse change? The meaning is not clear.

Reviewer 2 Report

The manuscript “Responses of Physiology, Photosynthesis and Related genes to Saline Stress in Cornus hongkongensis subsp. tonkinensis” by Yuan et al revealed that the moderate salt tolerance of C. tonkinensis mostly lies in a better photosynthetic system influenced by active photorespiration. The findings are interesting. However, some concerns should be addressed before this manuscript can be considered for publication. Hope the following comments be helpful to the authors.

(1)The preface mainly introduces the indicators of photosynthetic system, but the detection indicators of this study are not only focused on photosynthetic system, but also malondialdehyde (MDA), relative electrical conductivity (REC), and decreases in superoxide dispersion (SOD), photosynthetic rate (PN), stochastic conductivity (GS), and internal carbon dioxide concentration (CI), However, there is no introduction in the preface. You can briefly introduce the background information.

(2)In the part of materials and methods, the reagent used did not write the article number, but only the manufacturer's name.

(3)Pay attention to the logical relationship of the result part, which should be consistent with 2.2, 2.3, 2.4 and 2.5 of the material and method part.

(4)In the discussion part, it is suggested to make a prospect.

(5)References, 43, please pay attention to the page number. Please check other references carefully.

Round 2

Reviewer 1 Report

Many aspects appear to have been improved.
The following minor points must be revised.

Fig.S3, especially the protective mechanism of FNR depicted, needs to be redrawn to match the revised discussion.

The abbreviation FNR first appears in line 53, but its original wording is presented in line 55.
